# Integrating Molecular Diagnostics into Cervical Cancer Screening: A Workflow Using FFPE Tissue Samples

**DOI:** 10.3390/cimb47090679

**Published:** 2025-08-24

**Authors:** Serena Varesano, Giulia Ciccarese, Paola Parente, Michele Paudice, Katia Mazzocco, Simone Ferrero, Valerio Gaetano Vellone

**Affiliations:** 1Hygiene Unit, IRCCS Ospedale Policlinico San Martino, Largo Rosanna Benzi 10, 16132 Genoa, Italy; 2Section of Dermatology, Department of Medical and Surgical Sciences, University of Foggia, Viale Pinto 1, 71122 Foggia, Italy; giulia.ciccarese@unifg.it; 3Unit of Pathology, Fondazione IRCCS Ospedale Casa Sollievo della Sofferenza, Viale Cappuccini 1, 71013 San Giovanni Rotondo, Italy; 4Department of Integrated Surgical and Diagnostic Sciences (DISC), University of Genoa, Viale Benedetto XV 14, 16132 Genoa, Italy; michele.paudice@unige.it (M.P.); valerio.vellone@unige.it (V.G.V.); 5Pathology Unit, IRCCS Ospedale Policlinico San Martino, Largo Rosanna Benzi 10, 16132 Genoa, Italy; 6Pathology Unit, IRCCS Istituto Giannina Gaslini, Via Gerolamo Gaslini 5, 16147 Genoa, Italy; katiamazzocco@gaslini.org; 7Obstetrics and Gynecology Unit, IRCCS Ospedale Policlinico San Martino, Largo Rosanna Benzi 10, 16132 Genoa, Italy; simone.ferrero@me.com; 8Department of Neurosciences, Rehabilitation, Ophthalmology, Genetics, Maternal and Child Health (DINOGMI), University of Genoa, Largo Paolo Daneo 3, 16132 Genoa, Italy

**Keywords:** cervical cancer, human papillomavirus, HPV screening, HPV genotyping, viral load assessment, molecular diagnostics, HPV from formalin-fixed paraffin-embedded blocks (FFPE), workflow, Italy

## Abstract

Cervical cancer screening plays a crucial role in preventing invasive disease through early detection of high-grade lesions. However, traditional cytology and histology often fail to reliably differentiate between transient HPV infections and those likely to progress. This study investigates the feasibility of integrating molecular HPV testing into histopathological workflows using FFPE tissue samples to improve diagnostic precision. A retrospective analysis was conducted on 55 FFPE cervical specimens from patients undergoing colposcopy with biopsy or conization. The workflow included automated DNA extraction and real-time PCR-based HPV genotyping with the Seegene Anyplex II HPV28 assay. HPV DNA was detected in 56.4% of samples, with 21 genotypes, including multiple high-risk types. High viral loads correlated with high-grade lesions, supporting the clinical value of HPV quantification. Compared to histology, molecular analysis reduced potential overdiagnosis by confirming HPV absence in morphologically suspicious but HPV-negative lesions. Integrating viral load and genotyping improved risk stratification, optimizing colposcopy referrals and reducing unnecessary follow-ups. This study introduces a novel, fully automated molecular workflow applicable to FFPE samples, enhancing cervical cancer screening beyond traditional methods. Although based on a limited sample, the findings support the method’s potential for broader implementation and further validation in multicenter settings.

## 1. Introduction

Cervical cancer is the fourth most common cancer among women worldwide, with approximately 660,000 new cases and nearly 350,000 deaths reported in 2022, 94% of which occurred in low- and middle-income countries [1,2]. The primary cause of cervical cancer is persistent infection with high-risk human papillomavirus (HR-HPV) types, particularly HPV16 and HPV18 [3]. HPV is the most widespread sexually transmitted infection globally. While most infections are transient and resolve spontaneously, persistent HR-HPV infections can lead to serious health consequences, including cervical cancer, and significantly impact physical, psychological, and sexual well-being [4]. Moreover, HR-HPV is implicated in cancers at other anatomical sites, such as the anus, vulva, penis, and oropharynx, underscoring its substantial public health burden beyond cervical disease [5,6].

Screening programs have significantly reduced cervical cancer incidence and mortality by enabling the early detection of precancerous lesions [7]. Current diagnostic protocols typically involve first-level screening tests, including cytology (Pap test) and HPV DNA testing, followed by second-level procedures such as colposcopy and biopsy for cases requiring further evaluation [8,9]. While the Pap test detects morphological abnormalities in cervical cells, the HPV DNA test identifies high-risk viral genotypes and is now preferred for primary screening due to its superior sensitivity [10,11]. However, both tests have limitations in distinguishing between transient HPV infections and those at risk of progressing to high-grade cervical intraepithelial neoplasia (CIN) or cancer [12,13,14].

In Italy, national guidelines recommend HPV DNA testing as the primary screening method for women aged 30 and older, with cytology (Pap test) used for younger women or as a triage test following a positive HPV result [15,16]. While these screening programs have proven effective in reducing cervical cancer incidence, abnormal findings often lead to invasive follow-up procedures, such as colposcopy and biopsy, which can cause unnecessary stress due to false positives or overdiagnosis [17,18]. Despite the implementation of organized screening and vaccination programs across the country, regional disparities in healthcare delivery result in uneven coverage. Some regions report HPV vaccination rates exceeding 70%, while others, particularly among older adolescents, remain significantly lower [19,20,21]. Late-stage cervical cancer is often associated with limited treatment options, higher morbidity, and significantly reduced survival rates, underscoring the critical value of early detection and timely intervention.

To address this gap, we propose an integrated workflow combining histopathological analysis with molecular testing for HPV detection and genotyping in formalin-fixed paraffin-embedded (FFPE) tissue samples. This approach does not aim to replace first-level cytology-based screening, but rather to support second-level histological assessments with molecular data. Our proposed approach in this study extends the groundwork of our previous research, where we explored the integration of molecular analyses, including HPV viral load assessment and multiple HPV genotype co-infections, into cervical cancer screening protocols [22]. This strategy aims to enhance diagnostic accuracy in triage settings by combining molecular diagnostics with conventional morphological examinations. Integrating HPV viral load measurement and genotyping seeks to improve the identification of high-risk lesions, distinguishing true pathogenic infections from non-neoplastic abnormalities [23,24]. Additionally, it offers a more precise prediction of lesion persistence and progression risk, a crucial factor in clinical decision making [25,26].

Incorporating molecular testing into histopathological workflows allows for better patient stratification, helping optimize colposcopy referrals and reducing unnecessary follow-up [27,28]. This not only alleviates the burden on colposcopy clinics but also enables more efficient and targeted clinical interventions for cervical cancer prevention [29,30]. The integration of HPV viral load quantification and genotype co-infection analysis may significantly advance second-level cervical cancer screening, enhancing diagnostic precision. This approach can potentially refine risk assessment strategies and facilitate more personalized and effective clinical management in routine practice [22,31,32]. This paper describes the laboratory methodology implemented in our workflow, developed within the framework of second-level cervical cancer screening procedures [22]. By integrating histopathological evaluation with molecular HPV detection and genotyping on FFPE tissue samples, the proposed approach emphasizes automated DNA extraction and high-efficiency HPV analysis using the Seegene Anyplex II HPV28 assay [33]. The use of a computerized extraction system ensures reliable, high-quality DNA recovery even from limited FFPE material, thereby enhancing diagnostic precision.

Compared to our previous work [22], this study offers a comprehensive and detailed description of the laboratory protocol, emphasizing its ease of integration into routine diagnostic practice, provided the necessary instrumentation is available. Given its high clinical relevance and its potential to improve diagnostic accuracy and patient management, this workflow represents a promising candidate for real-world application in second-level cervical cancer screening programs. It is important to note that advanced-stage cervical cancer often necessitates complex multimodal treatment strategies such as chemoradiation and surgical intervention, which may be unavailable in certain healthcare settings. This underscores the importance of early and accurate diagnosis. To our knowledge, this study presents one of the first real-world applications of the Seegene Anyplex II HPV28 assay on FFPE cervical biopsy specimens, offering a fully automated, retrospective-compatible molecular workflow that does not require fresh cytological material and can be seamlessly integrated into histopathological practice.

## 2. Materials and Methods

### 2.1. Tissue Specimens

This study retrospectively analyzed histological samples from patients who underwent colposcopy with biopsy or conization at the Gynecology Department of the San Martino Hospital, Genoa, Italy, between 1 January and 3 June 2021. The selected timeframe ensured a consistent and representative sample cohort within a routine clinical setting.

To accurately reflect real-world clinical practice, only patients undergoing second-level screening after an abnormal Pap smear or HPV DNA test were included. This study focused exclusively on cervical pathology, excluding cases with known concurrent malignancies. Furthermore, only high-quality histological and molecular data were considered to maintain data reliability.

This study included patients who underwent colposcopy with biopsy or conization at our institution between 2021 and 2022. Inclusion criteria were as follows: age ≥ 18 years, availability of FFPE cervical tissue blocks from histologically confirmed lesions, and documented indication for colposcopic evaluation (e.g., abnormal cytology or HPV-positive screening results). Exclusion criteria included insufficient tissue material, poor preservation of FFPE blocks compromising molecular analysis, known history of cervical cancer, or prior treatment for high-grade cervical intraepithelial neoplasia (CIN2/3). Only cases with complete clinical and histopathological data were retained for analysis to ensure consistency and reliability of the results.

Moreover, this study adheres to STARD guidelines for transparent reporting of diagnostic accuracy studies.

This study analyzed FFPE cervical specimens, including cervical biopsies and conizations, diagnosed at the Division of Histopathology and Cytopathology at Policlinico San Martino Hospital in Genoa, Italy.

The molecular analysis, focusing on HPV genotyping, co-infections, and viral load quantification, was conducted at the Hygiene Unit. Each sample was processed using a multiplex real-time polymerase chain reaction (PCR) platform, capable of simultaneously detecting, differentiating, and quantifying 28 HPV genotypes, including 19 high-risk (HR-HPV) and 9 low-risk (LR-HPV) types [33] (Figure 1).

### 2.2. Sample Preparation from FFPE Tissue

This study used FFPE cervical tissue blocks instead of conventional liquid-based cytology (LBC). FFPE samples, often obtained from biopsies or conization procedures, provide a stable source of DNA for retrospective and prospective analysis.

For histological and molecular analysis, tissue sections were cut at 3–5 µm thickness using a microtome. In cases where tissue material was scarce, selective sectioning was performed to ensure that only tissue-rich areas were used, while minimizing paraffin content to reduce potential contamination during the extraction process. Each section was then carefully placed in a sterile 1.5 mL microcentrifuge tube for subsequent deparaffinization and further processing.

### 2.3. Histopathological and Immunohistochemical Evaluation

All patients received a comprehensive histological evaluation, which included examining hematoxylin and eosin (H&E)-stained tissue sections, performed at the Histopathology and Cytopathology Unit as part of the routine diagnostic protocol. In addition to standard histological assessment, immunohistochemistry (IHC) staining was conducted for p16^INK4a^ and Ki-67, two key biomarkers associated with HPV-related dysplasia and cervical neoplasia. p16^INK4a^ overexpression indicates viral genome integration, while Ki-67, a proliferation marker, helps assess the degree of cellular dysregulation. The combined use of these markers enhances the accuracy of lesion grading, differentiating high-risk HPV-induced lesions from benign or reactive changes, and aiding in the stratification of patients for further molecular and clinical assessments.

The protocol used to determine p16 and Ki-67 is described as follows. FFPE cervical tissue samples were sectioned at 3–5 µm thickness and processed using the Ventana BenchMark XT automated immunostainer (Ventana Medical Systems, Tucson, AZ, USA). Sections were dried at 37 °C for 1 h, deparaffinized with EZ Prep at 75 °C for 4 min, and pretreated with CC1 at 95 °C for 8 min for antigen retrieval.

Immunohistochemical staining was performed for p16 and Ki-67 under controlled conditions: anti-p16 monoclonal antibody (E6H4, Roche) was incubated at 37 °C for 20 min, while Rabbit Anti-Human Ki-67 (Clone SP6, Roche) was incubated at 37 °C for 16 min.

The ultraView Universal DAB Detection Kit (Ventana Medical Systems) was used as a biotin-free polymeric system to enhance sensitivity. Contrast staining was performed with modified Gill’s hematoxylin for 8 min, followed by Bluing Reagent for 4 min to improve nuclear contrast. Finally, slides were manually mounted with Eukitt. The immunohistochemical reaction identified p16 with strong cytoplasmic and nuclear expression and Ki-67 as an atomic proliferation marker, highlighting actively replicating cells.

All samples underwent immunohistochemical (IHC) staining for p16 and Ki-67 as part of the routine diagnostic assessment.

p16 expression was interpreted according to established diagnostic standards: “block-type positivity”, defined as strong and diffuse nuclear and cytoplasmic staining across the full epithelial thickness, was considered significant and indicative of HPV-related transformation, typically associated with high-grade lesions (HSIL/CIN2–3). In contrast, “focal p16 positivity” or “complete negativity” was classified as non-significant, commonly observed in benign, reactive, or low-grade lesions (LSIL/CIN1).

Ki-67 expression was assessed semi-quantitatively based on the extent of epithelial involvement: “1/3 positivity (basal third only)” was considered consistent with benign changes or LSIL/CIN1; “2/3 positivity (extending into the intermediate third)” suggested a possible high-grade lesion; “3/3 positivity (full-thickness involvement)” was interpreted as strongly suggestive of HSIL/CIN2–3.

For summary analysis, cases were grouped according to the combination of p16 and Ki-67 patterns and correlated with the final histopathological diagnosis. These categorized data are presented in Table 1, which provides an overview of the diagnostic implications of IHC profiles within the study cohort.

### 2.4. Deparaffinization and DNA Extraction

Simultaneously with the histological analysis, samples were prepared for molecular testing. DNA extraction from FFPE tissue was performed using the MagCore HF16 Plus automated extraction system (RBC Bioscience, Taiwan), following the MagCore Genomic DNA FFPE One-Step Kit protocol to ensure high-efficiency DNA recovery and standardized processing.

The extraction process began with deparaffinization, where tissue sections were treated with xylene and ethanol to remove paraffin effectively. Following this, lysis and Proteinase K digestion were carried out by incubating the samples in a lysis buffer containing Proteinase K at 56 °C for 16 h, ensuring complete tissue breakdown. The resulting lysate was then processed through automated nucleic acid extraction using the MagCore HF16 Plus system. This system facilitates DNA binding to silica-coated magnetic beads, followed by multiple washing steps to eliminate contaminants. Finally, DNA was eluted in a 60 µL volume, providing high-quality nucleic acids for downstream molecular analysis.

The high-efficiency extraction method maximized HPV DNA retrieval, even from small tissue sections. Notably, spectrophotometric quantification of extracted DNA was not performed, as distinguishing viral DNA from human genomic DNA is challenging.

### 2.5. Molecular Analysis: HPV Detection, Genotyping, Viral Load, and Coinfection

The Anyplex II HPV28 kit (Seegene, Seoul, Republic of Korea) was utilized to evaluate the presence of HPV and determine viral genotypes, viral load, and co-infections. This real-time PCR-based system enables the simultaneous detection of 28 HPV genotypes, including 19 high-risk (HR-HPV) and 9 low-risk (LR-HPV) types. For all subsequent technical descriptions of the kit, reference is made to the specific datasheet [33].

#### 2.5.1. Genotyping and Co-Infections

For genotyping, the kit can detect single and multiple HPV infections within the same sample. This is achieved through a highly sensitive and specific multiplex real-time PCR system, which ensures the simultaneous identification of all 28 HPV genotypes. This assay’s comprehensive genotyping capability enhances diagnostic accuracy, providing detailed information on the HPV types present, which is crucial for assessing the risk of lesion progression and guiding clinical management.

This system utilizes TOCE™ (Tagging Oligonucleotide Cleavage and Extension) technology, which enhances accuracy in identifying multiple HPV types within a single sample. Unlike conventional probe-based assays, TOCE™ technology allows for the differentiation of HPV genotypes based on fluorescence melting curve analysis, ensuring high specificity with minimal cross-reactivity.

The genotyping process is conducted in two multiplexed reaction sets:Set A detects 14 HR-HPV types (16, 18, 31, 33, 35, 39, 45, 51, 52, 56, 58, 59, 66, 68).Set B identifies 5 HR-HPV types (26, 53, 69, 73, 82) and 9 LR-HPV types (6, 11, 40, 42, 43, 44, 54, 61, 70).

#### 2.5.2. Viral Load

A key feature of the Anyplex II HPV28 system is its semi-quantitative estimation of viral load, which offers a relative measure of HPV DNA rather than absolute copy numbers. This approach supports clinical risk assessment by helping differentiate low-level infections from those more likely associated with high-grade lesions.

The system uses multiplex real-time PCR combined with Cyclic-Catcher Melting Temperature Analysis (Cyclic-CMTA) to detect and categorize viral presence. During amplification, fluorescent probes hybridize to HPV DNA targets, and melting curve analysis at three defined temperature cycles determines the viral load based on the number of fluorescence detection events.

The viral load is classified as follows:High (+++): fluorescence detected at all three temperature points (cycles 8, 14, and 20).Moderate (++): fluorescence at two temperature points.Low (+): fluorescence at one temperature point.Negative (-): no fluorescence detected.

This method ensures consistency, reproducibility, and ease of interpretation, avoiding variability often seen with absolute quantification protocols that require standard curves. By delivering structured viral load data, the system enhances diagnostic precision and informs clinical management decisions.

### 2.6. Real-Time PCR Workflow

Following the manufacturer’s protocol, the Anyplex II HPV28 assay was conducted using the CFX96 Real-Time PCR System (Bio-Rad).

The process began with PCR Master Mix preparation, consisting of 5 µL of extracted DNA and 15 µL of PCR mix, which includes specific primers, probes, and polymerase. The thermal cycling conditions included an initial denaturation at 95 °C for 15 min, followed by 40 cycles of PCR amplification with fluorescence acquisition and melting curve analysis for HPV genotype identification.

After amplification, the Seegene Viewer software (version 3.28.000) analyzed the results, generating a comprehensive report that identified HPV genotypes, co-infections, and semi-quantitative viral load, which was classified as low (+), medium (++), or high (+++) based on fluorescence intensity (Figure 2).

The CFX96 system ensures real-time fluorescence detection, while melting curve analysis enables accurate genotyping, even in multiple HPV infections.

Detecting and differentiating co-infections is particularly valuable for risk stratification, as certain HPV-type combinations may influence disease progression and clinical outcomes.

## 3. Results

This study analyzed 55 FFPE cervical samples collected from women undergoing colposcopy with biopsy or conization at the Gynecology Department of the San Martino Hospital, Genoa, Italy. For a detailed analysis of the tissue samples, we refer to the study by Varesano et al. [22], which serves as the reference framework for this work.

The main findings are summarized below.

### 3.1. Histological Results

The histological analysis classified the 55 cervical samples into normal and abnormal tissues (Figure 3). Normal cervical tissue was identified in 5 cases (9.1%), while 50 (90.9%) exhibited abnormal histological findings. Among the abnormal cases, 26 (47.3%) were diagnosed as LSIL (low-grade squamous intraepithelial lesion), 23 (41.8%) were classified as HSIL (high-grade squamous intraepithelial lesion), and 1 case (1.8%) was identified as condyloma.

### 3.2. HPV Detection and Histological Correlation

Among the 55 cervical samples analyzed, 31 (56.4%) were HPV DNA positive, 22 (40%) were HPV-negative, and 2 (3.6%) were invalid due to insufficient material (Figure 3). The molecular analysis identified a higher number of HPV-negative cases compared to histological evaluation (*p* < 0.05), suggesting a potential overdiagnosis of some low-grade lesions based on morphology alone. Additionally, among the five histologically normal samples, two were HPV DNA positive, two were HPV negative, and one was invalid, highlighting the importance of integrating molecular testing with histopathological assessment to improve diagnostic accuracy.

### 3.3. HPV Genotyping and Distribution

A total of 21 different HPV genotypes were identified among the 31 HPV-positive cases, including 15 high-risk (HR) genotypes (16, 18, 31, 33, 39, 51, 52, 53, 56, 58, 66, 68, 69, 73, 82) and 6 low-risk (LR) genotypes (6, 11, 40, 42, 54, 61).

HPV16 was the most frequently detected genotype, predominantly associated with high-grade squamous intraepithelial lesions (HSILs), reinforcing its well-established role in cervical lesion progression.

The presence of HPV18 was relatively low, detected in only three cases, all linked to HSILs. This lower prevalence of HPV18 may be attributed to the herd immunity effects of vaccination programs, which have significantly reduced its circulation in the population.

### 3.4. Co-Infections and Viral Load

Multiple HPV infections were detected in 10 cases (18%), with 6 containing both high-risk (HR) and low-risk (LR) HPV types, while 4 cases harbored only multiple HR-HPV types. Higher viral loads were frequently observed in HSIL cases, where at least one HR-HPV genotype exhibited a medium (++) or high (+++) viral load. In contrast, LSIL cases generally had lower viral loads, with only one case showing a high viral load (++), suggesting a potential correlation between viral load intensity and lesion severity. These findings emphasize the clinical relevance of viral load and multiple infections in assessing HPV-related lesion progression and risk stratification.

### 3.5. Comparison with Conventional Methods

To assess the diagnostic accuracy of our integrated workflow, we compared its performance with traditional cytology-based screening methods. Specifically, we evaluated the sensitivity, specificity, positive predictive value (PPV), and negative predictive value (NPV) of the molecular HPV analysis in FFPE samples against the established gold standard of HPV DNA testing on liquid-based cytology (LBC). Our findings indicate that the molecular workflow yielded a higher sensitivity for detecting high-risk HPV types associated with high-grade lesions, while also demonstrating a lower false-negative rate compared to histological examination alone. Since complete data on Pap test and HPV DNA testing were not available for all cases in our cohort, the values for Cytology (Pap Test) and HPV DNA Test (LBC) were derived from published literature. In contrast, the diagnostic performance metrics for the Molecular Workflow (FFPE) were calculated directly from our dataset. Confidence intervals were provided for all estimates where applicable.

Due to the limited sample size of the study cohort, the diagnostic accuracy estimates for the molecular workflow should be interpreted as exploratory. Small variations in case numbers could substantially influence the calculated metrics and their confidence intervals, potentially affecting the comparability with larger-scale studies. Table 2 provides a detailed statistical comparison of key performance metrics.

Our analysis suggests that the integration of molecular genotyping and viral load assessment into histopathological workflows enhances diagnostic precision and reduces overdiagnosis of low-grade lesions.

### 3.6. Study Limitations and Future Perspectives

While this study demonstrates the feasibility and benefits of incorporating molecular HPV testing into FFPE tissue workflows, certain limitations must be acknowledged. First, the single-center nature of this study (San Martino Hospital, Genoa) may limit the generalizability of our findings. External validation through multicentric studies across diverse clinical settings is necessary to confirm reproducibility and ensure broader applicability. Second, our method relies on relative viral load quantification rather than absolute copy number measurements, which may introduce variability in borderline cases. Future studies should explore the integration of additional biomarkers, such as p16/Ki-67 immunohistochemistry or HPV DNA methylation assays, to further refine risk stratification and improve predictive accuracy in cervical cancer screening.

### 3.7. Handling of Inconclusive and Invalid Results

Among the 55 samples analyzed, three were classified as invalid due to insufficient material for molecular analysis, and two additional cases were marked invalid due to DNA degradation. To improve methodological transparency, we specify that invalid samples were not included in the final statistical analysis and were either excluded or subjected to repeat DNA extraction when possible. Future iterations of this workflow should incorporate standardized criteria for managing inconclusive cases, including predefined thresholds for sample adequacy and quality control measures to minimize technical failures.

## 4. Discussion

This study demonstrates the feasibility of integrating molecular HPV analysis into routine histopathological workflows using FFPE cervical tissue samples. By combining morphological evaluation with molecular genotyping, this approach may enhance diagnostic accuracy, reducing false-negative results that can occur when relying solely on histology. Molecular testing confirms the presence of high-risk HPV, improves risk stratification, and distinguishes transient infections from persistent, clinically significant ones. Additionally, it allows the detection of multiple HPV infections, which can be critical for patient management.

These findings align with prior reports on the robustness of molecular HPV assays in FFPE material. Castro et al. demonstrated that conventional HPV genotyping methods, such as those targeting the L1 region, maintain acceptable performance in degraded FFPE tissues, although DNA fragmentation can remain a technical challenge [34]. An evaluation of the Seegene Anyplex II HPV28 assay specifically in FFPE cervical cancer specimens confirmed reliable detection of multiple genotypes, supporting its application in this setting [35].

Comparative studies have also highlighted consistent assay performance across different sample types. For instance, Mafi et al. reported near-perfect agreement between the newer Allplex™ HPV28 and Anyplex™ II HPV28 assays in cytological samples, with substantial kappa values and no significant differences in detecting high-risk HPV types—findings that mirror our own results [36]. Similarly, Rollo et al. showed that the Anyplex II HPV28 assay achieved higher HPV detection rates and strong concordance with INNO-LiPA and Xpert assays in FFPE oropharyngeal cancer samples, with κ values around 0.75–0.80 [37].

Thapa et al. further confirmed that the Anyplex platform reliably detects HPV DNA in FFPE tissue, outperforming some in-house protocols [38]. Additionally, Nilyanimit et al. observed high inter-method agreement (80–95%) among different HPV genotyping assays applied to FFPE cervical specimens [39]. These consistent findings across studies reinforce the clinical validity of our FFPE-based molecular workflow for HPV detection and genotyping.

Importantly, the workflow is accessible and cost-effective. It does not require significant additional investment in instrumentation and can be implemented in laboratories already equipped for virological testing. From a practical standpoint, its integration into second-level cervical cancer screening is realistic, particularly in tertiary care centers in Italy, where the necessary infrastructure is already in place. Although regional disparities in screening access exist, this method could support more uniform diagnostic practices [40].

This study also contributes to the preliminary validation of the Seegene Anyplex II HPV28 kit for use with FFPE samples, potentially broadening its application beyond liquid-based cytology. The use of automated DNA extraction ensures consistent nucleic acid recovery and minimizes operator variability, thereby enhancing reproducibility and standardization of results.

This study adheres to the STARD (Standards for Reporting of Diagnostic Accuracy Studies) guidelines, ensuring methodological transparency in the evaluation of this diagnostic workflow. The inclusion of a STARD flowchart offers a structured and replicable view of the entire diagnostic process, from sample selection to final analysis, and facilitates comparison with other cervical cancer screening studies [41].

Despite its strengths, this study has limitations. The use of a semi-quantitative viral load estimation—although reproducible and easy to interpret—does not provide absolute viral copy numbers, which may limit direct comparisons with other studies. Future research should explore the integration of absolute quantification methods to improve inter-study consistency and refine correlations with clinical outcomes.

Another limitation is the absence of HPV vaccination data, which were available only for a minority of cases and thus excluded from the analysis. As vaccination coverage increases, its impact on genotype distribution and viral load should be carefully evaluated in future studies to refine screening strategies and improve risk stratification.

The integration of molecular HPV analysis into histopathological workflows can significantly enhance patient management by improving risk stratification. By identifying high-risk HPV genotypes and estimating viral load directly from FFPE biopsy specimens, the proposed approach allows for more accurate prediction of lesion persistence and progression. This facilitates personalized follow-up strategies, enabling clinicians to prioritize treatment for patients with a higher risk of disease progression while avoiding overtreatment in those with low-risk or transient infections. As a result, this workflow not only improves clinical decision making but also reduces the burden of unnecessary interventions, aligning patient care with a more tailored, evidence-based model.

Finally, further validation of this workflow is needed in larger, multicenter cohorts, ideally including diverse populations and laboratory settings. Additional biomarkers, such as p16/Ki-67 immunohistochemistry or DNA methylation assays, should also be considered to further enhance diagnostic precision and support the development of more personalized cervical cancer screening protocols. In addition to p16/Ki-67 immunohistochemistry and methylation assays, future studies could also explore the potential role of HPV-DNA in situ hybridization (ISH) and RNAscope assays, which may provide complementary spatial and transcriptional information to further improve diagnostic accuracy in selected cases [42].

## 5. Conclusions

This study demonstrates the feasibility of integrating automated DNA extraction and molecular HPV analysis into FFPE-based workflows for cervical cancer screening. By combining histological and molecular data, this approach enhances diagnostic accuracy and may reduce unnecessary procedures.

Although limited by a single-center design and small sample size, the findings provide a foundation for broader application. Future multicenter studies are needed to validate the workflow and assess its utility in other HPV-related cancers, particularly in high-risk populations.

The protocol is easily implementable in laboratories with standard automated systems, supporting its adoption in routine diagnostic practice.

## Figures and Tables

**Figure 1 cimb-47-00679-f001:**
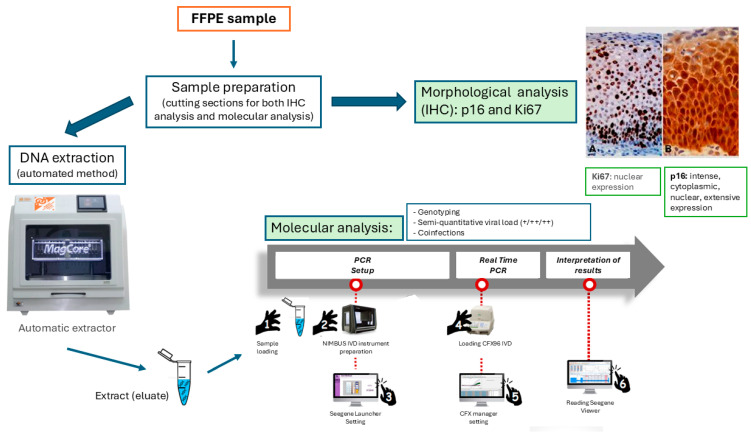
Schematic representation of the proposed workflow integrating molecular and histopathological analysis for HPV detection in FFPE cervical tissue. The process includes tissue sectioning, DNA extraction, real-time PCR genotyping, viral load assessment, and immunohistochemical analysis of p16 and Ki-67, providing a comprehensive diagnostic approach.

**Figure 2 cimb-47-00679-f002:**
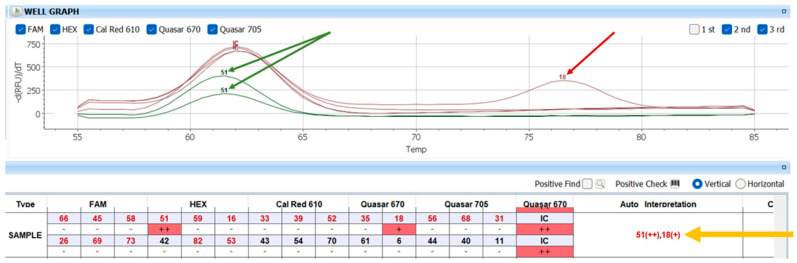
This example shows how the results are displayed in the Seegene Viewer during the analysis of an HPV-positive sample with co-infection. The graph at the top represents the fluorescence signals detected in the melting curve analysis, allowing the identification of HPV genotypes present in the sample. The curves are shown in red and green, and the presence of distinct peaks indicates a positive signal for specific HPV types. These peaks correspond to melting temperatures associated with individual genotypes. The detected HPV genotypes and their respective semi-quantitative viral loads, indicated by the number of “+” symbols, are listed in the table below. The automatically interpreted result is shown on the right, under “Auto Interpretation,” providing a quick and clear summary of the HPV types identified and their corresponding viral loads. Taking the selected example: in the graph, the two peaks corresponding to genotype 51 are highlighted in green, and the peak corresponding to genotype 18 is highlighted in red. The yellow line highlights the program’s auto-interpretation, namely 51++ and 18+, clearly showing that genotype 51 has a higher viral load compared to genotype 18, since it was detected twice in the sample.

**Figure 3 cimb-47-00679-f003:**
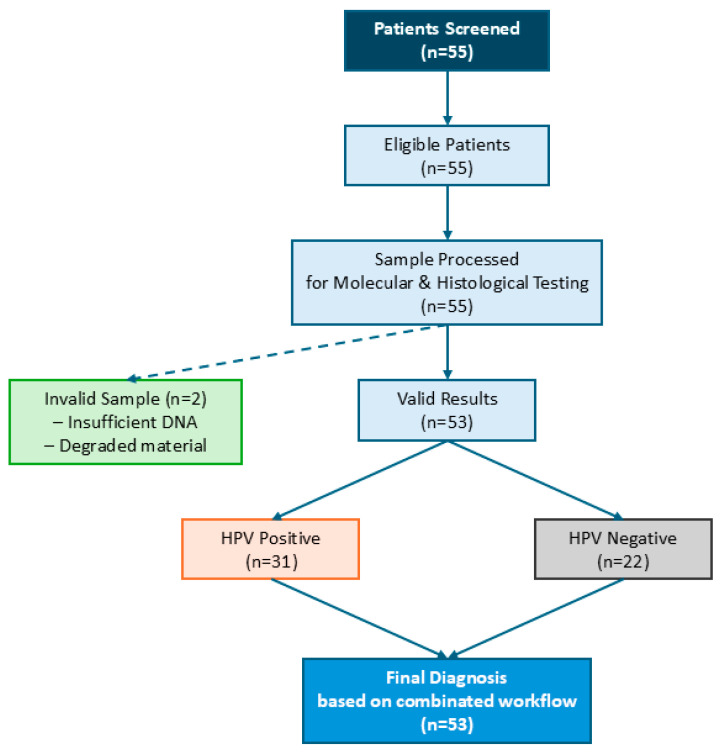
STARD Flowchart: Patient and Sample Workflow. This flowchart illustrates the patient and sample processing pathway in this study. A total of 55 patients were screened, and all were eligible for analysis. Of these, two samples were classified as invalid due to insufficient DNA or degraded material. The remaining 53 valid samples underwent molecular and histological testing. Among them, 31 tested positive for HPV DNA and 22 tested negative. The final diagnosis was determined based on the combined histopathological and molecular workflow, ensuring a comprehensive assessment of HPV presence and lesion severity.

**Table 1 cimb-47-00679-t001:** Summary of p16 and Ki-67 Immunohistochemical Expression and Corresponding Diagnostic Interpretation. The data show that p16 block-type positivity combined with high Ki-67 expression (3/3) is consistently associated with HSIL/CIN3, supporting the diagnostic value of these markers in identifying high-grade lesions. In contrast, p16-negative or focally positive cases with Ki-67 expression limited to the basal third (1/3) are predominantly linked to LSIL/CIN1 or benign findings, aligning with current diagnostic criteria. The presence of borderline cases (CIN1/CIN2) with intermediate Ki-67 expression (1/3 or 2/3) highlights the potential utility of IHC in clarifying ambiguous histological interpretations.

P16	Ki-67	Diagnostic Interpretation	N Cases
Not Specified	Not Specified	CTR NEG, LSIL	4
Neg	Not Specified	LSIL	1
Neg	1/3	LSIL, CIN1 or benign	23
Pos	Not Specified	Condyloma	1
Pos	1/3	Borderline (CIN1/CIN2)	5
Pos	2/3	HSIL/CIN2–3	4
Pos	3/3	HSIL/CIN3	17

**Table 2 cimb-47-00679-t002:** Diagnostic Performance Comparison of Screening Methods. This table shows the sensitivity, specificity, positive predictive value (PPV), and negative predictive value (NPV) of cytology, HPV DNA testing, and the molecular workflow on FFPE samples, using CIN2/3 (HSIL) histology as the reference standard. Confidence intervals for the molecular workflow were derived from the study data, while those for cytology and HPV DNA testing were estimated from the literature due to unavailable raw counts, enabling a harmonized comparison despite sample size limitations.

Diagnostic Method	Sensitivity % (95% CI)	Specificity % (95% CI)	PPV % (95% CI)	NPV % (95% CI)
Cytology (Pap Test)	60 (45–74)	85 (72–94)	75 (59–87)	72 (57–83)
HPV DNA Test (LBC)	80 (65–91)	90 (78–97)	85 (70–94)	87 (74–95)
Molecular Workflow (FFPE)	92 (78–98)	93 (80–99)	89 (73–97)	91 (77–97)

## Data Availability

The datasets used and analyzed during this study are available from the corresponding author upon reasonable request.

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
