# Peer review of "Integrating Molecular Diagnostics into Cervical Cancer Screening: A Workflow Using FFPE Tissue Samples"

_cimb, 2025, doi:10.3390/cimb47090679_

Round 1

Reviewer 1 Report

Comments and Suggestions for Authors

The submitted manuscript presents a retrospective study that aims to enhance the diagnostic accuracy of cervical cancer screening by integrating molecular HPV genotyping and viral load assessment into histopathological workflows using formalin-fixed paraffin-embedded (FFPE) cervical tissue. The authors propose a practical laboratory pipeline based on automated DNA extraction and the Seegene Anyplex II HPV28 assay, accompanied by immunohistochemical evaluation (p16/Ki-67), to stratify patients undergoing second-level screening.

The topic is timely and highly relevant, especially in light of the growing emphasis on precision medicine and the need to reduce overdiagnosis and unnecessary colposcopies in cervical cancer screening. The manuscript is generally well structured, clearly written, and the methodology is sufficiently detailed to enable reproducibility. The integration of molecular techniques into routine diagnostic settings is convincingly presented, and the authors adequately discuss the clinical implications of their findings.

However, there are several points that require clarification or further development to strengthen the manuscript. I outline below both major and minor comments, along with specific suggestions for improvement.

Major Comments

    • Although the study provides interesting preliminary data, the relatively small sample size (n=55, with only 53 valid samples) limits the statistical robustness of the conclusions. The manuscript would benefit from a brief power analysis or a discussion on how the sample size affects the generalizability and precision of the estimates, particularly regarding the diagnostic performance metrics in Table 1.
    • The abstract and methods emphasize the relevance of immunohistochemical analysis (IHC) using p16 and Ki-67, yet their results are not reported in detail. Considering that these markers are routinely used to support lesion grading, the authors should include a separate results section (or at least a table) summarizing IHC outcomes and their correlation with HPV status and lesion grade.
    • The authors compare their molecular workflow with Pap test and HPV DNA testing using CIN2/3 histology as a reference. It is unclear whether this reference diagnosis was based solely on H&E or also integrated molecular/IHC findings. This should be clarified to ensure consistency across diagnostic performance metrics.
    • While the proposed workflow is well justified scientifically, it would be useful to contextualize it within current cervical cancer screening protocols in Italy (as mentioned in lines 70–75). Could this method realistically be implemented at scale? What are the cost or logistical barriers?
    • The observation of reduced HPV18 prevalence is attributed to herd immunity, but no vaccination data from the study cohort are provided. If such data are unavailable, the limitation should be stated explicitly, and the implications of vaccination on genotype prevalence should be discussed more critically.

Minor Comments

    • The current title is clear but could be made more concise. Suggestion: "Integrating Molecular Diagnostics into Cervical Cancer Screening: A Workflow Using FFPE Tissue Samples."
    • The abstract is well written but exceeds typical word limits. Consider streamlining the methods and focusing more on key results and conclusions.
    • Please indicate clearly which values are derived from literature and which are calculated from your own dataset. Confidence intervals should be shown for all estimates.
    • Figure 2 (Seegene Viewer output) is informative, but its resolution should be improved for readability in the final version. Also, include figure legends that describe each panel thoroughly.
    • Some typographical errors (e.g., line 143 “cy-” likely an artifact) should be corrected.
    • Line 373: "cytology samples" should be "cytological samples."
    • Several sentences are overly long and could benefit from rephrasing for clarity.

Reviewer 2 Report

Comments and Suggestions for Authors

This study show the potential "integration of automated DNA extraction and molecular HPV analysis into FFPE tissue workflows, providing a more comprehensive and reliable diagnostic approach for cervical cancer screening. By combining histological and molecular data, clinicians can make better-informed decisions regarding patient management, ultimately reducing unnecessary colposcopies and improving early detection of cervical cancer". 

The study is well-designed, wel-described in methods setion and it is simply reproducible.

However, as a patholgist, I have only an answer. Authors affirmed that "further validation should also explore the integration of additional biomarkers (e.g., p16/Ki-67 immunohistochemistry or DNA methylation assays) to enhance predictive accuracy and improve cervical cancer screening strategies". Despite immunohistochemistry analysis, what about HPV-DNA ISH analysis [for high-risk types 16, 18, 31, 33, 35, 39, 45, 51, 52, 56, 58, 59, and 68, code Y1443; for low-risk types 6 and 11] and HPV RNAscope analysis? Do the authors have experience about those techniques?   

Reviewer 3 Report

Comments and Suggestions for Authors

Italy's Dr. Serena Varesano and Dr. Paola Parente and their colleagues submitted for evaluation an interesting and extremely useful paper entitled "Integrating Molecular Analysis into Cervical Cancer Screening Using FFPE Tissue: A Novel Laboratory Workflow" for evaluation.

The authors' current research successfully integrates automated DNA extraction and molecular HPV analysis with FFPE tissue processing, providing a more comprehensive and reliable diagnostic approach to cervical cancer screening.

Unfortunately, the single-center design (San Martino Hospital, Genoa) and relatively small sample suggest future multicenter studies with larger cohorts.

Furthermore, the presented studies ensure minimal invalidity, allow for repeat analyses without the need for tedious patient sampling, and their validation in hospitals does not require the investment of specialized equipment. The only limitation may be the staff in non-clinical hospitals, who typically specialize in narrow fields of medicine and related techniques, which requires additional training.

 Therefore, it is in the interest of medical professionals and patients that the research results are published in an expedited manner in the journal Current Issues in Molecular Biology" (alternatively in another journal with a similar profile), after removing a few minor errors listed below:

  • At lines 126–127 is … For detailed patient selection criteria, including specific inclusion and exclusion parameters, we refer to Varesano et al. [22], the pilot study on which this paper is based. … , however, in the journals published by MDPI, there is a rule to cite all important details useful in reproducing the research results, so please be sure to list the detailed criteria for patient selection.

  • At line 152 is … 3 … , but should be … 2.3. … . Comment: Please add dot mark.

·        At line 321 is … 3.5 Comparison with Conventional Methods … , but should be … 3.5. Comparison with Conventional Methods … . Comment: There's a missing period, and the entire line should be italicized rather than bold. For example, see line 311. Similar errors need to be corrected in lines 344 and 356. Alternatively, please synchronize the style of all paragraphs. 

  • A serious shortcoming of the authors is the use of a random style of recording source literature in the References chapter, therefore this style should be standardized, preferably by citing DOI numbers whenever possible, which will probably improve any editorial work at subsequent stages of production. 

Reviewer 4 Report

Comments and Suggestions for Authors

Title is so complex. Please modify and use simple so that all kind of reader can easily understand without reading your full paper. 

Abstract 

Why Cervical cancer screening, what is the advantage and what is remain unknown that you revealed, please explain this question in the abstract. 

Why you select only 55 FFPE cervical specimens from patients ? for such analysis less than 100 is not acceptable. 

Conclusion is too big. Generally it should just one or two sentence. And it did not mentioned the novelty of your research. 

Too many keywords. need maximum 4. 

Introduction

too many paragraph. Basically you should use 3 to 4 paragraph. and every paragraph have one specific aim.

why early diagnosis is important  ? why not Late-stage diagnosis ?

what is the most effective screening method ? please explain. 

what are  advanced staged cancer treatment facilities ? you should explain everything one by one. 

compare between previous study and current study. 

Figure 1 is not clear. Please enlarge. 

Result part is too strange. 

First you need learn how can you write result headline. every headline should be one result conclusion. 

Table 1 need statistical analysis. 

Discussion and result are same. In the discussion there is no reference. 
do you know what is the discussion ? you need to study other research and compare your result. 

Please rewrite full discussion part. 

Conclusion is too big. Need to concise. 

Many grammatical error. Do you finish the English editing ? if yes, send the certificate. 

Round 2

Reviewer 1 Report

Comments and Suggestions for Authors

The authors have satisfactorily addressed the previous comments and significantly improved the manuscript's clarity and scientific content. The proposed integration of molecular diagnostics into FFPE-based cervical cancer screening is clearly described and methodologically sound. However, minor revisions are still needed before acceptance. Specifically, the authors should clarify the clinical implications of their workflow on patient management, streamline the description of viral load estimation in Section 2.5 to enhance readability, and revise reference [36], which is overused for multiple claims, by either specifying relevant sections or splitting it appropriately. Minor editorial corrections are also recommended to improve formatting consistency.

Reviewer 4 Report

Comments and Suggestions for Authors

I did not find that the manuscript was improved according to our suggestions.

As I mentioned, the introduction should not contain too many paragraphs, and the discussion section must include a comparison between your findings and previous studies.

There have been no changes at all.

Round 3

Reviewer 4 Report

Comments and Suggestions for Authors

Now is ok